# Quality improvement exercises in Inflammatory Bowel Disease (IBD) services: A scoping review

Katie Ridsdale[1]*, Kajal Khurana[1], Azizat Temidayo Taslim[2], Jessica K. Robinson[2], Faith Solanke[2], Wei Shao Tung[2], Elena Sheldon[1], Daniel Hind[1], Alan J. Lobo[3]

1 School of Health and Related Research, The University of Sheffield, Sheffield, United Kingdom, 2 Medical School, The University of Sheffield, Sheffield, United Kingdom, 3 Sheffield Inflammatory Bowel Disease Centre, Sheffield Teaching Hospitals NHS Foundation Trust, Sheffield, United Kingdom

* k.ridsdale@sheffield.ac.uk

**Data Availability Statement:** All relevant data are within the manuscript and its Supporting Information files.

## Abstract

### Objective

Quality Improvement initiatives aim to improve care in Inflammatory Bowel Disease (IBD). These address a range of aspects of care including adherence to published guidelines. The objectives of this review were to document the scope and quality of published quality improvement initiatives in IBD, highlight successful interventions and the outcomes achieved.

### Design/method

We searched MEDLINE, EMBASE, CINAHL and Web of Science. Two reviewers independently screened and extracted data. We included peer reviewed articles or conference proceedings reporting initiatives intended to improve the quality of IBD care, with both baseline and prospectively collected follow-up data. Initiatives were categorised based on problems, interventions and outcomes. We used the Quality Improvement Minimum Quality Criteria Set instrument to appraise articles. We mapped the focus of the articles to the six domains of the IBD standards.

### Results

100 studies were identified (35 full text; 65 conference abstracts). Many focused on vaccination, medication, screening, or meeting multiple quality measures. Common interventions included provider education, the development of new service protocols, or enhancements to the electronic medical records. Studies principally focused on areas covered by the IBD standards 'ongoing care' and 'the IBD service', with less focus on standards 'pre-diagnosis', 'newly diagnosed', 'flare management', 'surgery' or 'inpatient care'.

### Conclusion

Good quality evidence exists on approaches to improve the quality of a narrow range of IBD service functions, but there are many topic areas with little or no published quality

**Funding:** The author(s) received no specific funding for this work.

**Competing interests:** I have read the journal's policy and the authors of this manuscript have the following competing interests: Professor Alan Lobo has acted as a consultant and advisory board member for Takeda Pharma, Janssen and Bristol Myers Squibb. This does not alter our adherence to PLOS ONE policies on sharing data and materials.

improvement initiatives. We highlight successful quality improvement interventions and offer recommendations to improve reporting of future studies.

## Introduction

Inflammatory Bowel Disease (IBD), which includes ulcerative colitis (UC) and Crohn's disease (CD), is characterised by inflammation of the gastrointestinal system, significant morbidity, and lifelong medication [1]. In 2017, 84.3 per 100,000 persons suffered from IBD worldwide [2]. Annual healthcare costs in Europe are over £10,000 per prevalent CD case, and over £6000 per prevalent UC case [3]. Guidelines promote optimum healthcare based on medical research findings and expert opinions. For example, The IBD standards [4] were created by IBD UK, a partnership of 17 patient and professional organisations. They outline how care should be high-quality for all patients with IBD, at all stages of their journey, including how care should be organised and managed to achieve this.

However, development of guidelines does not necessarily result in their implementation; guideline adherence is often reported as low. For example, despite guidelines on immunisations for IBD patients, low vaccination rates have been observed in both adult and paediatric groups who receive immunosuppressive treatment [5–7].

Quality Improvement (QI) "attempts to change clinician behaviour and, through those changes, lead to improved patient outcomes" [8], across domains including safety, effectiveness, patient-centredness, timeliness, efficiency, and equity [9]. Quality improvement projects are often local and time-limited [10]. Systematic overviews can help improve planning quality improvement projects by summarising what is already known and where there are evidence deficits [11]. A recent systematic review focused on publications addressing US quality metrics for adult IBD [12]. It found that successful approaches empowered non-physicians, targeted multidisciplinary teams, prompted clinicians using Electronic Medical Records (EMR) and restructured care delivery. Here, we present a broader scoping review of published quality improvement studies from any setting which address IBD care for adults or children.

Scoping reviews can be used to categorise available evidence on a topic and summarise knowledge gaps [13]. In this study, our objectives were to: 1) Characterise published reports of QI interventions in IBD care; 2) Identify high quality, successful studies and their intervention components; and 3) Map the current literature to the IBD Standards, and identify knowledge gaps.

## Methods

### Protocol

This scoping review was conducted using standard procedures and is reported in accordance with the "Preferred Reporting Items for Systematic Reviews and Meta-Analyses extension for Scoping Reviews" (PRISMA-ScR) statement (S1 Checklist). The protocol was fixed at the beginning of March 2022 (S1 Protocol).

### Eligibility criteria

Eligible studies: a) were peer reviewed articles or conference abstracts published between 2009 and 2023 in the English language; b) included people of any age or gender with CD or UC; c) described at least one initiative aimed at assisting, facilitating, or improving the quality of care;

and d) retrospective or prospectively collected baselines and prospectively longitudinal outcome data. Studies were considered ineligible if they were based exclusively on retrospective or cross-sectional data; were based in mixed populations who may not all have IBD (e.g., colorectal surgery); just discussed the process of forming an initiative without gathering data about the project itself; were health economic studies; were review papers; or any studies involving non-human participants. If there were numerous records for a single study, the most comprehensive dataset from that record was used. There were no restrictions on the study setting (e.g. level of healthcare, and urban/rural), location, or country of origin.

### Information sources and search

On 2nd June 2023 we searched MEDLINE, EMBASE, CINAHL and Web of Science, with no date restrictions, using thesaurus and free-text terms related to IBD, CD, UC and QI (S1 Appendix). We also searched Google Scholar and reference lists of included studies.

### Selection of sources of evidence

The search results were uploaded to Rayyan [14] and duplicates removed. The title and abstract of each paper were screened for eligibility by at least two reviewers, with disagreements settled by a third.

### Data charting and process

Data collection forms were developed, piloted and iteratively modified in Microsoft Excel. The final form reflected domains of the Quality Improvement Minimum Quality Criteria Set (QI-MQCS) instrument [15] and other, topic-specific, fields (see data items). Authors were not contacted for missing data due to the number of studies and time constraints.

### Data items

- Study characteristics: country, population (adult/paediatric), number of patients, number of centres, study design (guided by the QI-MQCS handbook [15]).

- The following categories were inductively derived by the team:

  ◦ *Problem* (studies were coded to one category): Preventive Health (sub categories: Vaccination; Screening; VTE Prophylaxis; Mental Health; Pre-treatment Tests); Health Maintenance (sub categories: Medication; Drug Monitoring; Endoscopic Scoring; Enteral therapy; Patient/Family Guidance; Follow-up; Pre-visit Planning Compliance; Treat to Target; Stratified Care; Urgent Care/Triaging; Time to Treatment; Care Co-ordination; Transitional Care); Multiple Quality Measures; Clinician Workload; Clinician Guideline Knowledge; and Costs/Resources.

  ◦ *Intervention* (studies could be coded to more than one category): Provider Education; New Protocol; New Documentation; Informatics; Reminder System; Patient Education; Accountability; and Team Change (e.g. Changes to team structure or composition).

  ◦ *Outcome* (studies could be coded to more than one category): Process outcomes: Protocol/Documentation Adherence; Treatment Rate; Vaccination Rate; Testing Rate; Screening Rate; Steroid Use; Follow-up Rate; Patient Knowledge; and Costs. Clinical outcomes: Remission Rate; Disease Activity; Length of Stay; Patient Satisfaction; Adverse Events; Nutritional Status; Admissions; Readmission; Relapse; Quality of Life; and Pain.

- The stage of the patient journey targeted by the improvement initiative, categorised using the IBD Standards UK system [4]. The standards are separated into 7 sections, which are summarised below. These are further broken down into a number of statements, which can be access on the IBD UK website.

1. The IBD service: A well-organised and managed local Inflammatory Bowel Disease IBD service is necessary to provide safe, consistent, high quality, personalised care.

2. Pre-diagnosis: Early and accurate diagnoses allows for treatment and support sooner, and better management.

3. Newly diagnosed: The right treatment and support should be in place for newly diagnosed patients.

4. Flare Management: Patients should recognise a flare, and access the right specialist advice and treatment to manage it as quickly as possible.

5. Surgery: Surgery should be timely, led by surgeons with the right expertise, and with effective multidisciplinary working. Patients should fully understand their options and be offered psychological support.

6. Inpatient Care: Inpatients should be admitted to a specialist ward with appropriate facilities, should receive a holistic assessment, and be given clear information on discharge.

7. Ongoing care: As IBD is a fluctuating, lifelong condition, people need ongoing care, including a personalised care plan.

### Critical appraisal

At least two reviewers applied the 16-item QI-MQCS instrument [15] to each study, with discrepancies resolved by a third reviewer. Scores were divided into terciles: the four lowest scores (two to five) were ranked as low quality; the five middle scores (six to ten) were ranked as moderate quality, and the four highest scores (eleven to fourteen) were ranked as high quality.

### Synthesis

Narrative/tabular summaries and frequency counts were generated for study characteristics, problem, intervention and outcome categories, critical appraisal domains and knowledge gaps. A tabular synthesis was also produced detailing theories of change for high quality studies, including a description of their overarching themes.

## Results

### Study selection

After the elimination of duplicates, 1515 titles and abstracts were evaluated for eligibility; 1385 were excluded. On review of 148 full texts, 100 studies were included: 35 peer reviewed publications [16–50] and 65 conference abstracts [51–115]. Fig 1 details reasons for rejection. We also searched Google Scholar, and citations from included reports, but all were duplicates of already retrieved articles or did not meet inclusion criteria.

### Characteristics of sources of evidence

Of the 100 included studies (see references and more information in S1 Dataset), the country was unknown in five cases; the remainder were conducted in the US (n = 77), the UK (n = 11),

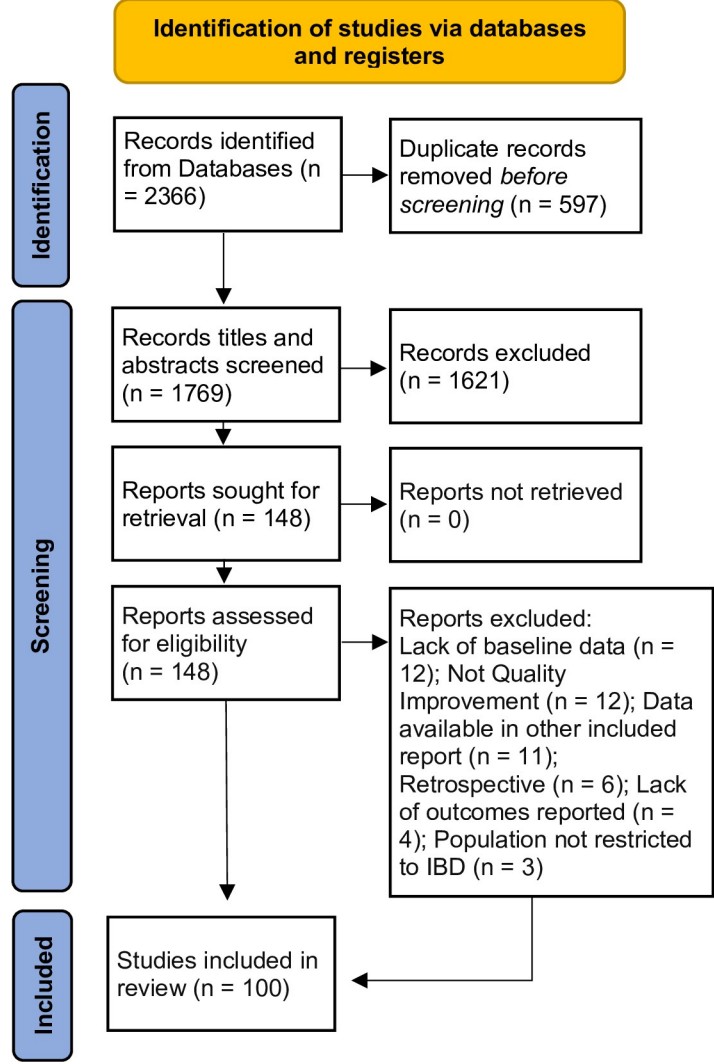

**Fig 1. Flow diagram illustrating study identification and exclusion in this review as recommended by PRISMA SCR [116].**

Israel (n = 2), Singapore (n = 2), Australia (n = 1), Belgium (n = 1), and Italy (n = 1). Twenty-six reports involved adults, 44 were on a paediatric population, and the population was unclear in 30. Seventy-two studies were single centre; 10 did not state the number of centres. Other studies ranged from 2 to 38 centres. Study designs were 'pre-post' (n = 92); time series (n = 4); observational cohorts (n = 2); individually-randomised (n = 1) and cluster-randomised (n = 1) controlled trials.

## Problem/intervention/outcome categories

The quantity of papers which address each problem, intervention and outcome category are shown in Table 1. The problems addressed are broken down into adult and paediatric studies in Fig 2. Process outcomes were reported 124 times, whereas clinical outcomes were reported 50 times. Sixty-three studies reported only process outcomes.

**Table 1. Quantity of identified Inflammatory Bowel Disease quality improvement papers (n = 100) categorised into problems, interventions and outcomes.**

| Problem | N | Intervention | N | Outcome | N |
|---|---|---|---|---|---|
| Multiple quality measures | 21 | New Protocol | 51 | *Process outcomes (n = 124)* | |
| Vaccination | 15 | Provider Education | 47 | Protocol/Documentation Adherence | 30 |
| Screening* | 12 | Informatics | 44 | Treatment Rate | 21 |
| Medication | 11 | Reminder System | 19 | Screening Rate | 19 |
| Drug Monitoring | 6 | New Documentation | 18 | Vaccination Rate | 18 |
| VTE Prophylaxis | 5 | Patient Education | 15 | Testing Rate | 13 |
| Enteral Therapy | 3 | Accountability | 7 | Steroid Use | 8 |
| Follow-up | 3 | Team Change | 4 | Follow-up Rate | 7 |
| Patient/Family Guidance | 3 | | | Costs | 5 |
| Urgent Care/Triaging | 3 | | | Patient Knowledge | 3 |
| Mental Health | 2 | | | *Clinical outcomes (n = 50)* | |
| Pre-treatment tests | 2 | | | Adverse Events | 10 |
| Stratified Care** | 2 | | | Remission Rate | 9 |
| Endoscopic Scoring | 2 | | | Disease Activity | 6 |
| Time to Treatment | 2 | | | Admissions | 6 |
| Clinician Workload | 2 | | | Length of Stay | 6 |
| Pre-Visit Planning Compliance | 1 | | | Patient Satisfaction | 5 |
| Treat to Target | 1 | | | Nutritional Status | 3 |
| Transitional Care | 1 | | | Readmission | 2 |
| Care Co-ordination | 1 | | | Relapse | 1 |
| Clinician Guideline Knowledge | 1 | | | Quality of Life | 1 |
| Costs/Resources | 1 | | | Pain | 1 |

Each paper was categorised into only one problem, but may be assigned multiple intervention/outcome categories.

*Screening included bone health (n = 4), *Clostridioides difficile* (n = 2), anaemia (n = 2), vitamin D (n = 1), nutrition (n = 1) iron deficiency (n = 1), and social determinants of health (n = 1).

**Stratified care included primary sclerosing cholangitis (n = 1) and obesity (n = 1).

## Critical appraisal of included studies

Most full-paper reports in this review (32 out of 35) were rated between 10 and 14 on the QI-MQCS (Fig 3). Research only available as conference abstracts scored between 2 and 10, due to reporting limitations.

The following items were reported well in the 35 full-text articles: Organisational Motivation (n = 35), Data Source (n = 35), Timing (n = 35), Limitations (n = 34), Intervention Rationale (n = 33) Intervention Description (n = 32), Organisational Characteristics (n = 31), and Implementation (n = 30). Other items were reported more poorly: Organisational Readiness (n = 27), Adherence/Fidelity (n = 26), Sustainability (n = 24), Comparator (n = 22), Study Design (n = 16), Spread (n = 16), Health Outcomes (n = 12), Penetration/reach (n = 7).

The five lowest reporting items for the 65 conference abstracts were: Health Outcomes (n = 13), Comparator (n = 11), Organisational Readiness (n = 7), Spread (n = 7), Limitations (n = 3), and Penetration/Reach (n = 3).

## Knowledge gaps

When mapping the studies to the areas of improvement defined by the IBD Standards [4], studies most commonly focussed on problems relating to 'The IBD service' (n = 41) or 'Ongoing care' (n = 40). There were gaps in QI research into the areas of 'Newly Diagnosed Patients'

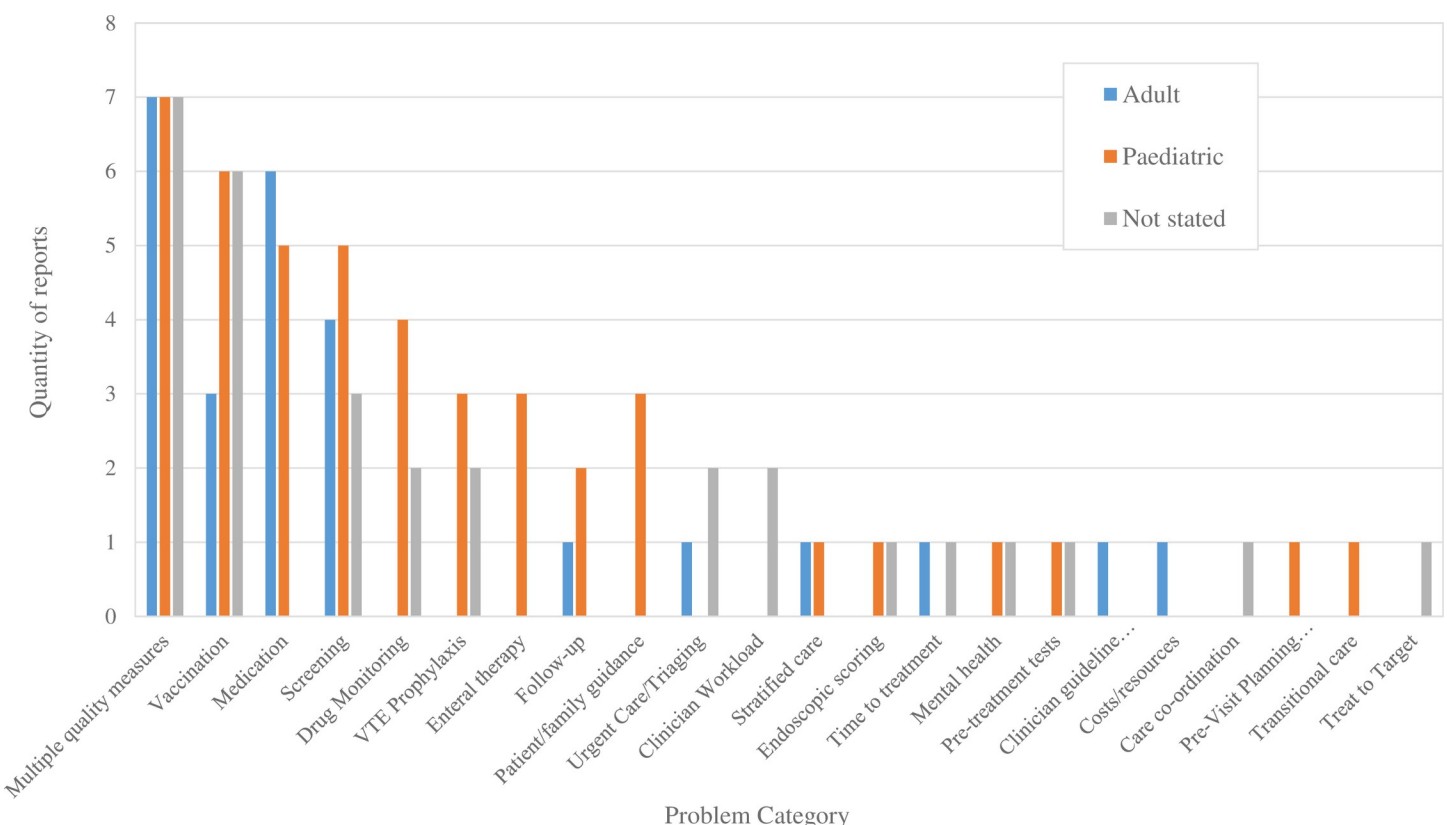

**Fig 2. The categorised problems addressed by Inflammatory Bowel Disease quality improvement reports (n = 100), separated into adult/paediatric populations.** 26 studies reported an adult population, 44 reported a paediatric population, and the population was not stated in 30 studies.

(n = 10), 'Inpatient Care' (n = 9) and 'Flare Management' (n = 8), and a distinct paucity of research into 'Pre-Diagnosis' (n = 1) and 'Surgery' (n = 1). 9 papers did not map to any areas defined by the IBD Standards.

## Description of initiatives

All 30 papers that were rated as 11 or higher during critical appraisal (all of which were full papers) reported successful initiatives. This included papers focussing on the following problems:

- Preventive Health: Vaccination, and Screening (Bone health, nutrition, anaemia and *C. difficile)*;

- Health Maintenance: Medication, Drug Monitoring, Enteral Therapy, Follow up, and Pre-visit Planning Compliance;

- Adherence to multiple quality measures (process and health outcome measures chosen as quality indicators, derived from American College of Gastroenterology guidelines [17], physician quality reporting system performance measures [18, 25], National Quality Strategy priorities [26], European Crohn's and Colitis Organisation guidelines, or discussions between paediatric IBD centre representatives, policy makers and administrators [50]).

- Clinician Guideline Knowledge;

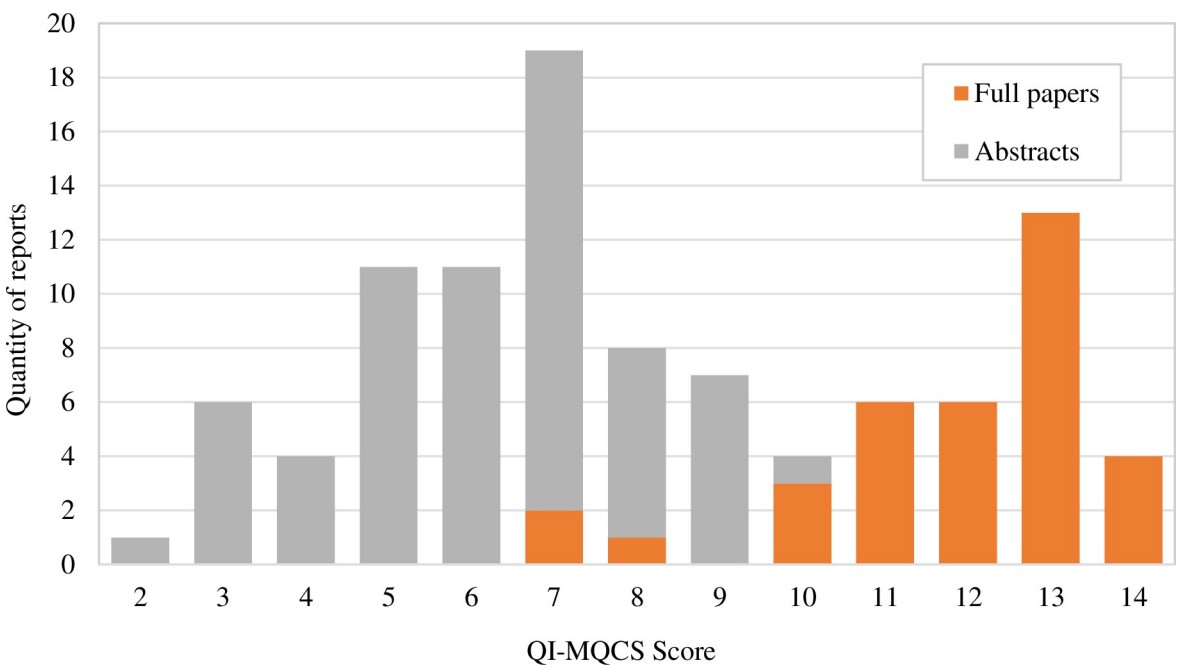

**Fig 3. QI-MQCS [15] score for published Inflammatory Bowel Disease qualit0y improvement reports included in this review (n = 100).** Reports are split into full texts (n = 35) and conference abstracts (n = 65). Maximum possible score is 16.

These 30 articles could be grouped into 3 overarching themes: Workload and workflow; Guidelines and standards; and Education and Information. Details of the intervention components and outcomes of the 30 'high quality' papers are included in Table 2. Intervention components and outcomes of all 35 full texts and 65 conference abstracts can be found in S1 Dataset.

## Discussion

We performed an extensive search of the published literature on QI initiatives in IBD, screened a large quantity of articles and identified 100 studies across a variety of settings, between 2010 and 2023. Studies covered a range of problem areas but some areas are underrepresented. High quality studies successfully demonstrated improvements through a variety of interventions. Conference abstracts provide useful insights but the lack of information hinders their replication. Intervention success was measured using a wide variety of outcome measures; more frequently in process outcomes than clinical outcomes. This may be unsurprising; though clinical outcomes are ultimately the QI target for improvement, process outcomes may give a more direct indication of the impact of the QI initiative, as they are less likely to be influenced by external factors [117].

### Implications for patients, clinicians and policy makers

QI is a 'bottom-up' process that strongly relies on regional norms and the knowledge base of local clinicians. Despite efforts to standardise care, there still appears to be variation in clinical practice, much of which is unwarranted [118]. This review complements the more focused work of Fudman and colleagues [12] in offering a broader evidence base for improvement strategies in IBD. From that review and our own, there is now a sound evidence base for improvement approaches to several problems. Some interventions were complex, such as formation of a new rapid access clinic [24] or creating new job roles [12, 23], but many studies

**Table 2. Intervention components and outcomes of Inflammatory Bowel Disease quality improvement reports rated high quality by the QI-MQCS (n = 30), organised by problem category.**

| Problem category | Reference | Overarching theme | Intervention components | Outcomes |
|---|---|---|---|---|
| Pre-visit Planning compliance | Dykes et al 2017 [21] | Guidelines and standards | Summarised evidence into guidelines; Checklist tested on sample patients; Automated form pulling data from EMR. | Patients receiving complete bundle increased from 0% to ~100%; ~70% now receiving pneumococcal vaccine. |
| Clinician Guideline Knowledge | Weizman et al, 2021 [49] | Education and information | Educational iPad video for hospitalised patients summarising inpatient management guidelines; accessible throughout hospitalisation. | Higher trust in physician at discharge and 6 months in intervention group (p<0.05); higher satisfaction at discharge in intervention group (p<0.05) but not sustained at 6 months; more tuberculosis skin testing within 48 hours in (p = 0.013); no significant differences in length of stay or colectomy. |
| Clinician Workload | Ewelukwa et al, 2018 [23] | Workload and workflow | Scribes assisted with documentation during clinic visits; Posters and discussions raised awareness of scribe clinics. | Patient satisfaction increased from 6.8 to 9.2 (p<0.01); Appointment length decreased by 13.5 minutes (p<0.05); Revenue increased 536% over scribe salary. |
| Costs/resources | Fofaria et al, 2019 [24] | Workload and workflow | Patient questionnaires to gauge interest in out of hours clinics and telephone monitoring; Information campaign on benefits of telephone clinics using posters for patients and posters/meetings for clinicians and emergency department staff. | Percentage of eligible patients transferred to telephone clinics increased from 17.6% to 59.3% using in-clinic discussion method. Patient satisfaction scores remained high and non-inferior to baseline scores. |
| Drug Monitoring | Guido et al, 2020 [28] | Guidelines and standards | Established local standard of care; created database; infliximab therapy plans built into EMR to automatically trigger lab orders; email reminder system; updated pre-visit planning process to manually review patients and updated templates. | Post-induction therapeutic drug monitoring increased from 43% to >80% (p<0.001); infliximab therapeutic drug monitoring increased from 59% to 82% (p<0.01); 36% of levels <5 μg/mL. |
| | Hellmann et al, 2021 [30] | Guidelines and standards | Designed care algorithm for parent education on infliximab doses; education for providers; updated pre-visit planning process to review patients weekly and feedback to whether providers adhered to the guideline; best practice alerts in EMR to recommend dose adjustment for low levels. | Percentage of patients with level ≥5 μg/mL and checked in 12 months improved from 73% to 80% in 6 months (p<0.01) and 88% at 1 year; sustained remission improved from 62% to 75%. |
| | Kelly et al, 2019 [33] | Guidelines and standards | Created Standard Operating Procedure (SOP) for paediatric infliximab infusions; Pre-infusion safety checklist; Education sessions for nurses on SOP and checklist; Checklist and SOP refined based on feedback. | Median safety checklist completion increased from 46% to 81% in one unit (P<0.05) and 91% to 95% in another unit (P<0.05); Laboratory screening adherence increased from 81.8% to 95.2% (P<0.001). |
| Enteral therapy | Shaikhkhalil et al, 2018 [46] | Guidelines and standards | Standard Exclusive Enteral Nutrition algorithm developed and iteratively refined; calorie/fluid table to determine formula needs; education session for providers; talking points for discussions with patients/families; weekly rounds to review patients. | Exclusive Enteral Nutrition utilisation increased from <5% to ~50% (p<0.01); 71% of patients completing ≥8 weeks of achieved remission; in patients completing there were significant reductions in disease activity. |
| Follow-up | Prendaj et al, 2019 [41] | Workload and workflow | Research coordinator called patients monthly to schedule appointments; made clinic appointments for infliximab patients on the same day as their infusion; educated physicians on recommended visit frequency and educated patients that guidelines recommend visits twice a year. | Median documented visits within 200 days increased from 64% to 83% (p<0.0001). Increase sustained for 1 year. |
| | Savarino et al, 2016 [43] | Workload and workflow | Pre-visit planning for patients with upcoming visits; team made care recommendations for patients with active disease; then expanded over cycles to include patients with mild/moderate disease and more providers. | Clinical remission rate increased from 77% to 83% over 12 months. 78% of providers found recommendations helpful. |
| | Choe et al, 2021 [20] | Workload and workflow | Implemented urgent scheduling slots for IBD patients with wait times >2 weeks; dedicated IBD clinic scheduler booked follow-ups; provided education to inpatient gastrointestinal team on contacting scheduler for patients needing timely follow-up | Mean wait time decreased from 40.4 days to 21.9 days after intervention, but change not statistically significant (p = 0.408). Only two responses to patient satisfaction survey limited interpretation. |
| Medication | Gupta et al, 2019 [29] | Workload and workflow | Offered external infusion options; created standard order sets to facilitate ordering process; established post-infusion nurse communication protocols; educated providers and patients on external infusion options. | Significant increase in patients offered external infusions (7% to 48%, p<0.0001); Significant increase in patients receiving external infusions (7% to 30%, p<0.0001). |
| | Kaimakliotis et al, 2021 [32] | Education and information | Brief educational lecture and pocket guide on stepwise approach to analgesia provided to internal medicine and emergency medicine residents; re-education every 3 months. | Inpatient opioid use decreased from 43.4 mg to 7.7 mg morphine equivalents (p<0.01); discharge prescriptions decreased from 3.7% to 0% (p = 0.03); Length of stay decreased from 5.3 to 3.7 days (p<0.04); 90-day readmissions decreased from 25% to 7% (p<0.04; there was no significant difference in pain scores. |

*(Continued)*

**Table 2.** (Continued)

| Problem category | Reference | Overarching theme | Intervention components | Outcomes |
|---|---|---|---|---|
| | Morris et al, 2022 [38] | Education and information | Education on biosimilars for providers, including presentations, information sheets, process map and frequently asked questions sheets; for inpatients, a clinical pharmacist was consulted before patient started on IV anti-Tumour Necrosis Factor drugs to allow them to review preferred product on insurance plan; enrolled eligible patients in co-payment assistance programs. | Biosimilar utilisation increased from 1% to 96% (p<0.001); estimated cost savings of $381,000 (average sales price) and $651,000 (wholesale acquisition cost) over 20 months; No significant difference in clinical outcomes. |
| | Sandberg et al, 2019 [42] | Workload and workflow | Created standardised order set to simplify infusion orders; transitioned eligible patients to 60 min rapid infusions after initial doses; eliminated 30 min post-infusion observation time; pharmacy installed pass-through window to improve efficiency of infusion preparation. | Average door-to-door time decreased from 279 min to 151 min; estimated 128 min per patient freed per year; no adverse events with rapid infusions. |
| | Kozlicki et al, 2023 [34] | Workload and workflow | Dashboard integrated the electronic health record and pharmacy claims database to identify patients on biologics needing updated labs before next refill due; speciality pharmacists reviewed dashboard and messaged nurses via electronic health record if patient labs were required; messages were sent 4 weeks in advance | Frequency of treatment gaps decreased from 80% to 32%; median gap length decreased from 21 days to 11 days. |
| | Ong et al, 2022 [39] | Workload and workflow | Educational tutorials, information leaflets and workflow map to guide physicians on eligibility criteria; implemented accelerated 1 hour infliximab infusion protocol for eligible patients; shifted infliximab collection from distant pharmacy to nearby pharmacy. | Mean infliximab infusion time reduced by 47% (142min to 75min, p<0.001); total time spent in infusion centre reduced by 52% (214min to 106min, p<0.001); 3 mild infusion reactions out of 152 infusions (2%). |
| **Multiple quality measures** | Bensinger et al, 2019 [18] | Workload and workflow | Implemented a note template with prepopulated sections for Physician Quality Reporting System measures; order set with one-click access to vaccines, bone density scans, tuberculosis, hepatitis B testing; patient education handout on vaccinations, bone health, cancer screening, tobacco cessation added to after-visit summary provided to patient. | Significant increases in documentation rates of influenza immunization (19–59%, p<0.001), pneumococcal immunizations (2–38%, p<0.001), tobacco cessation (28.6–77.8%, p = 0.049); sustained improvements at 1 year. |
| | Greene et al, 2015 [25] | Education and information | Four accredited educational activities for physicians: web-based private audit feedback session, two interactive online videos and a 20-page monograph on quality improvement and evidence based approaches. | No significant differences for overall adherence to any of the Physician Quality Reporting System IBD quality measures, but low performing gastroenterologists showed significantly greater improvement on adherence to 4 measures after education compared to high performers. |
| | Yogev et al, 2021 [50] | Guidelines and standards | 21 Israeli paediatric IBD centres participated in national quality improvement program with monthly anonymous feedback on performance; position papers published mid-study on use of faecal calprotectin and anti-TNF levels. | Significant increase in: obtaining anti-TNF levels (66% to 87%, p = 0.005); faecal calprotectin utilization (63% to 71%, p = 0.008); bone density testing (53% to 68%, p = 0.002). Significant improvement in: calprotectin <300 mg/mg (60% to 66%, p = 0.015); composite endpoint of inflammation resolution (36% to 53%, p = 0.007). |
| | Benjamin et al, 2023 [17] | Workload and workflow | Patient questionnaire to educate on increased risk of vaccine-preventable illnesses, skin cancers, osteoporosis, cervical cancer and mental health problems, and identify deficiencies; paper order sheet for health maintenance orders, entered into EMR after visit; case management log to track patients seen, orders placed, and patient follow-through. | Ordered health maintenance item increased from 20% to 100%; completion rate unchanged; mean patient engagement score increased from 3.0 to 4.6. |
| **Screening** | Smith et al, 2023 [48] | Guidelines and standards | Education of gastrointestinal division on iron deficiency/ iron deficiency anaemia (ID/IDA) algorithm created by IBD team & haematologist; copies of algorithm placed in clinical spaces & EMR; iron studies added to diagnostic checklist used by IBD nurse coordinators; EMR smart tools created to facilitate ordering iron studies. | Screening rates increased from 20% to >90% (p<0.001); Of those screened, 88% had ID/IDA; 77% with ID/IDA treated within 30 days. |
| | Lambl et al, 2019 [35] | Guidelines and standards | Daily cleaning with bleach and ultra violet terminal room disinfection; guidelines and order set changes to restrict clindamycin and fluoroquinolones; modified testing algorithm to reduce inappropriate polymerase chain reaction tests. | *C. difficile* rate declined 55.5% (p = 0.002); high-risk antibiotic use declined 88.1% (p<0.001); antibiotic restrictions associated with 20.6% infection decline. |
| | Gold et al, 2022 [27] | Guidelines and standards | Implemented malnutrition screening program using modified Malnutrition Universal Screening Tool; recommendations for high risk patients; smart tool and smart phrase built into EMR. | Significant increase in number screened (3% to 63%, p<0.01); significant increase in micronutrient testing for high risk patients (0% to 63%, p<0.01); small non-significant increase in dietician referrals (33% to 37%, p = 0.9). |

*(Continued)*

**Table 2.** (Continued)

| Problem category | Reference | Overarching theme | Intervention components | Outcomes |
|---|---|---|---|---|
| | Shah-Khan et al, 2019 [45] | Education and information | Education lecture for providers; flyer summarising guidelines posted in provider area; EMR prompt for providers to consider ordering bone mineral density screening. | Bone mineral density screening rate significantly increased from 10.8% to 81.8% (p<0.01). |
| | Breton et al, 2021 [19] | Guidelines and standards | Multidisciplinary team created paediatric evidence-based care pathway for ID/IDA; EMR dashboard created tracking anaemia screening, iron deficiency screening, iron supplementation; providers received individualised monthly reports from dashboard data. | Iron deficiency screening increased from 31.7% to 63.6%; treatment rates increased from 38.2% to 49.9%; anaemia prevalence decreased 35.8% to 29.7% (p = 0.003). |
| **Urgent care/ triaging** | Melmed et al, 2021 [37] | Workload and workflow | Formed multidisciplinary care teams; reserved urgent slots in clinics; proactive communication with high-risk patients; educated patients on seeking urgent care. | System wide improvement in multiple measures: need for urgent care, hospitalisations, CT scan utilisation, steroid use, opioid use (p<0.05) (18–50% relative reductions). |
| **Vaccination** | Parker et al, 2013 [40] | Education and information | Education form on needed vaccines given to patients; vaccines offered and given at same visit by nurse. | Influenza vaccination increased from 54% to 81% (p<0.001); pneumococcal vaccination increased from 31% to 54% (p<0.001). |
| | Shores et al, 2019 [47] | Guidelines and standards | Implemented customised EMR prompts requiring action on influenza vaccination; educated providers on using prompts and importance of vaccination; had providers demonstrate prompt use in second year. | Documented vaccination improved from 10% to 39% in year 1 and 61% in year 2 (p<0.001); vaccine counselling for unvaccinated patients improved from 27% to 77% by year 2 (p<0.001). |
| | McNicol et al, 2022 [36] | Guidelines and standards | Created standardised 3-dose hepatitis B vaccine protocol and workflow for clinic and infusion centre; nurse training and competency assessment on vaccine; pre-visit planning to order vaccines and serologies in EHR before visits; engaged multi-disciplinary stakeholders through meetings and trainings. | Proportion of eligible patients who received hepatitis B vaccine dose 1 increased from 7% to 100%, sustained for over 12 months (p<0.05); proportion of patients who completed 3-dose vaccine series increased from 0% to 82%; 92% demonstrated hepatitis B seroprotection after 3 doses. |

All reports in this table scored 11–14 using the QI-MQCS tool. See S1 Dataset for details on reports which scored below 11. EMR = Electronic Medical Record; IBD = Inflammatory Bowel Disease.

employed less time and resource intensive approaches, such as educational sessions. Common interventions included: patient educational handouts; different forms of learning sessions aimed at providers; posters or flyers in clinics; improvement of the pre-visit planning processes; standardised order sets; enhancements to the EMR system; and updates to, or development of new, decision aids and guidelines. Most of the published interventions were carried out in single centres, therefore to ensure the findings can be replicated, centres should consider specific differences between published studies and their own units [10]. High quality studies are identified in Table 2, with interventions which could be incorporated directly into services. These fall into overarching themes describing guidelines and standards, workload and workflow, and education and information. Services may also gain insights and ideas for QI by reviewing the remaining studies listed in the S1 Dataset; though there may be less detail published on these projects.

Different interventions were successfully implemented targeting similar problem areas. For example, Parker. et al [40] developed a vaccination update paper form with educational information for patients, which improved influenza vaccination rates by 27% and pneumococcal vaccination rates by 23%. Shores. et al [47] improved documented paediatric influenza vaccination rates by 51% by implementing a new EMR system that allowed prompts and automated letters to patients, and educating nurses and providers about this system and the importance of documentation. Guido. et al [28] improved therapeutic drug monitoring of post-induction anti-TNF levels from 43% to over 80% in under a year and sustained this for a further year, by establishing a local standard of care, creating a database, implementing therapy plants into the EMR, developing a reliable reminder system for order entry and follow-up, and updating the pre-visit planning process. Hellmann. et al [30] increased the proportion of infliximab infusion

plans that had an appropriate drug level rechecked from 61% to 83% in one year through PDSA cycles involving education, development of an algorithm, feedback from providers, and EPIC software enhancements to flag abnormal levels and automate rechecking.

## How to describe the pattern of attention in included studies?

Studies from paediatric centres account for just under half the studies included and extend the breadth of areas of practice addressed in studies included in this review. Categories were more restricted in studies which stated an adult population, though the spread is similar if studies which did not state their population are assumed to be on adults. The intervention categories we have used are by necessity broad. Therefore, although a particular overarching area may be represented, important facets may not. For example, approaches to improve vaccination practice are represented from adult and paediatric centres but important implications relating to use of Janus kinase inhibitors are not. Studies most frequently address adherence to multiple quality measures, vaccination and medication. Those relating to medication frequently address issues of biologic medication and its administration. The rapid proliferation of new agents in IBD has not been reflected in quality improvement studies to examine their optimum use.

There are few initiatives focussed on the 'pre-diagnosis' or 'surgery' phase of the patient journey, as well as underrepresentation of the newly diagnosed, urgent care, and inpatient care phases. Interestingly, key themes where high quality studies were not identified include those addressing patient experience and what matters to individual patients, team structure and appropriateness of drug–or surgical–treatment; for example the timing or sequencing of such treatments. Given pressures on in-patient and out-patient services, changes that impact on clinician workload are scarcely addressed.

## Recommendations for future QI research

Non-publication of QI studies, publication in abstract form, or inadequate reporting in peer reviewed articles, all represent a substantial barrier to the dissemination of initiatives demonstrating good practice–and therefore their adoption. During critical appraisal, full text papers regularly failed to report on: the proportion of eligible units who participated (Penetration/Reach); health related outcomes; the potential, tools, or evidence of rollout to other units (Spread); the study design; and the sustainability or potential for sustainability (for example by reference to organisational resources or policy changes required). Although it is a critical appraisal tool, we recommend the use of the QI-MQCS, to supplement the Standards for Quality Improvement Reporting Excellence (SQUIRE) checklist [119], when reporting improvement initiatives. This would improve reporting of context-specific conditions that might affect knowledge translation to other settings, improving the ability to replicate findings at other centres.

Significant gaps have been identified, where new work is needed and which should meet standards for high quality studies. We recommend the development of improvement initiatives based on geographically-specific guidelines and formal IBD care quality criteria [120].

## Limitations

The main limitation of our review is that it is restricted to published improvement initiatives, which might not be representative of all QI initiatives. Our use of Fan and colleagues' [8] definition of quality improvement restricted our scope to exercises targeting clinicians which reported baseline and prospective follow-up data. Furthermore, our search strategy may have missed published articles that did not use QI terminology, but were by our definition eligible QI studies. The inclusion of studies published in abstract was an attempt to increase included

studies, but Fig 3 demonstrates that the assessed quality of such studies is less than those published in full. In addition, there is inevitably some overlap between problem categories, whereby an intervention might fit into more than one. We dealt with this by allocating the category with best fit.

## Conclusions

Good quality evidence exists for approaches to improving the quality of some IBD service functions, but this addresses only a narrow range and there are many topic areas with little or no published quality improvement initiatives. Successful interventions have been made but a wide range of areas are not represented. Use of the QI-MQCS to supplement the SQUIRE checklist would improve reporting of future studies.

## Supporting information

**S1 Checklist. Preferred Reporting Items for Systematic reviews and Meta-Analyses extension for Scoping Reviews (PRISMA-ScR) checklist.**
(DOCX)

**S1 Protocol. Scoping review protocol (fixed in March 2022).**
(DOCX)

**S1 Appendix. Search strategy for scoping review on quality improvement initiatives in Inflammatory Bowel Diseases.**
(DOCX)

**S1 Dataset. Published Inflammatory Bowel Disease quality improvement studies identified in scoping review (n = 100).** Abbreviations: US = United States; UK = United Kingdom; Paed = Paediatric; N/S = Not Stated; NA = Not Applicable; IBD = Inflammatory Bowel Disease; QI-MQCS = Quality Improvement Minimum Quality Criteria Set (critical appraisal tool); EMR = Electronic Medical Record. *Study is only available as conference abstract, not full text report.
(DOCX)

## Author Contributions

**Conceptualization:** Daniel Hind, Alan J. Lobo.

**Investigation:** Katie Ridsdale, Kajal Khurana, Azizat Temidayo Taslim, Jessica K. Robinson, Faith Solanke, Wei Shao Tung, Elena Sheldon, Daniel Hind.

**Project administration:** Katie Ridsdale, Kajal Khurana, Daniel Hind.

**Supervision:** Katie Ridsdale, Elena Sheldon, Daniel Hind, Alan J. Lobo.

**Visualization:** Katie Ridsdale.

**Writing – original draft:** Katie Ridsdale, Kajal Khurana.

**Writing – review & editing:** Katie Ridsdale, Kajal Khurana, Azizat Temidayo Taslim, Jessica K. Robinson, Faith Solanke, Wei Shao Tung, Elena Sheldon, Daniel Hind, Alan J. Lobo.

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
