## [Decision Letter · Decision Letter 0]

15 Dec 2023

PONE-D-23-29768Quality Improvement Exercises in Inflammatory Bowel Disease (IBD) Services: A Scoping ReviewPLOS ONE

Dear Dr. Ridsdale,

Thank you for submitting your manuscript to PLOS ONE. After careful consideration, we feel that it has merit but does not fully meet PLOS ONE’s publication criteria as it currently stands. Therefore, we invite you to submit a revised version of the manuscript that addresses the points raised during the review process.

We look forward to receiving your revised manuscript.

Kind regards,

Valérie Pittet, PhD

Academic Editor

PLOS ONE

[I have read the journal's policy and the authors of this manuscript have the following competing interests: Professor Alan Lobo has acted as a consultant and advisory board member for Takeda Pharma, Janssen and Bristol Myers Squibb.]. 

Reviewers' comments:

Reviewer's Responses to Questions

**Comments to the Author**

1. Is the manuscript technically sound, and do the data support the conclusions?

Reviewer #1: Yes

2. Has the statistical analysis been performed appropriately and rigorously? 

Reviewer #1: N/A

3. Have the authors made all data underlying the findings in their manuscript fully available?

Reviewer #1: Yes

4. Is the manuscript presented in an intelligible fashion and written in standard English?

Reviewer #1: Yes

5. Review Comments to the Author

Reviewer #1: The authors present a very thorough scoping review of published quality improvement interventions in IBD. This expands on a prior work to include pediatric studies and non-American studies, as well as a broader variety of metrics. Bringing all of these studies together is an achievement and worthy of publication. However, the synthesis and "take home messages" from the review could be improved. Most notably, Table 2 needs significant improvement, and the outcomes of unsuccessful interventions need mention.

--The authors refer to the IBD standards, which presumably are the NHS standards. These are likely unfamiliar to most outside the UK. Consider reframing the categorizations to a set of topics that is more universally understood. Alternatively, though less ideally, these should be explained in adequate detail for a reader unfamiliar with the concept to understand their use in the paper.

-- Stratified care is referred to - could the authors clarify why the topics of anemia, PSC and obesity are grouped together under this?

-- The results describe successful interventions, but not those which reported a lack of success. Publication bias notwithstanding, it is worthwhile to understand what hasn't work (rather than just what has).

-- Table 2 is the most important piece of the work, however it is too vague to be helpful to readers looking to understand characteristics of successful interventions. Consider structuring the table such that each intervention category under each problem category has a row. This would allow the measures used to be reported in a new column, and a more detailed description of the intervention for each can be provided. Most crucially, it should be clear exactly what is being improved and how this was accomplished. This table should serve almost as a "standalone" such that a reader would understand much of the paper looking at just this table, though of course should be more digestible than S4

-- S4 should be expanded similarly so that it is clear what is being improved and how, rather than just general categories. The present version is helpful to narrow the selection of papers to look at if interested in a specific metric, but more summary is appropriate.

-- There is a potential missed opportunity in the discussion to translate the results into take home messages for clinicians looking to improve the quality of their center's care. Consider making thematic conclusions about what has worked -- either by type of metric or by type of intervention.

-- A limitation of the work is the search strategy, which may have missed works that were not captured by QI terms, but that were in fact QI studies targeting specific metrics. This is understandable and should be noted as a limitation.

6. PLOS authors have the option to publish the peer review history of their article (what does this mean?). If published, this will include your full peer review and any attached files.

Reviewer #1: No

---

## [Author Response · Author response to Decision Letter 0]

10 Jan 2024

We would like to express our thanks to you and the reviewer for reviewing our manuscript. We appreciate the consideration of our paper, and the detailed response provided. The comments given were insightful, and by addressing these we believe that we have improved the quality of this report. Please see details on how we have addressed the reviewer comments below:

• Use of the IBD standards: The IBD standards are not NHS standards, but were developed by IBD UK. We believe these are useful for categorisation as they describe the stages of the patient journey, allowing us to highlight areas of the journey with fewer published quality improvement papers. However, we agree that the definition of these required more explanation, therefore have defined these in the introduction (page 3) and further expanded on this in the methods (pages 7 to 8).

• Grouping of anaemia, PSC and obesity in stratified care: PSC and obesity have been grouped here as these two papers focus on specific subsets of IBD patients – screening/treatment specific to those with additional conditions. However, after consideration, the anaemia article has been re-categorised into ‘Screening’ as this did focus on the whole population. 

• Description of unsuccessful studies: As mentioned by the reviewer, due to publication biases, almost all articles report some level of success, leaving little insight into unsuccessful interventions. As we have now added results to Table 2 and S4, it should be clearer how successful the individual interventions were.

• Expansion of Table 2: We have added more details on the intervention components and the outcomes/results for all papers. Though we attempted to structure as suggested, with intervention categories in separate rows alongside outcomes, this looked very unclear, as individual articles used multiple interventions. However, by adding the details, and by adding an extra ‘overarching theme’ column, we believe this more clearly shows what is being improved and how it was accomplished, allowing the table to serve as a standalone if necessary.

• Expansion of S4: Similarly to table 2, we have added intervention components and outcomes to S4 to provide more summaries of the research.

• Translation of results into take-home messages: We have added overarching themes to Table 2, and added further thoughts to the discussion.

• Limitation of search strategy: We have added this the limitations section. 

Finally, as requested, I have updated our statement regarding competing interests, as follows. Thank you for changing this in our online submission form: I have read the journal's policy and the authors of this manuscript have the following competing interests: Professor Alan Lobo has acted as a consultant and advisory board member for Takeda Pharma, Janssen and Bristol Myers Squibb. This does not alter our adherence to PLOS ONE policies on sharing data and materials.

---

## [Decision Letter · Decision Letter 1]

24 Jan 2024

Quality Improvement Exercises in Inflammatory Bowel Disease (IBD) Services: A Scoping Review

PONE-D-23-29768R1

Dear Dr. Ridsdale,

We’re pleased to inform you that your manuscript has been judged scientifically suitable for publication and will be formally accepted for publication once it meets all outstanding technical requirements.

Kind regards,

Valérie Pittet, PhD

Academic Editor

PLOS ONE

Additional Editor Comments (optional):

Reviewers' comments:

Reviewer's Responses to Questions

**Comments to the Author**

1. If the authors have adequately addressed your comments raised in a previous round of review and you feel that this manuscript is now acceptable for publication, you may indicate that here to bypass the “Comments to the Author” section, enter your conflict of interest statement in the “Confidential to Editor” section, and submit your "Accept" recommendation.

Reviewer #1: All comments have been addressed

2. Is the manuscript technically sound, and do the data support the conclusions?

Reviewer #1: Yes

3. Has the statistical analysis been performed appropriately and rigorously? 

Reviewer #1: N/A

4. Have the authors made all data underlying the findings in their manuscript fully available?

Reviewer #1: Yes

5. Is the manuscript presented in an intelligible fashion and written in standard English?

Reviewer #1: Yes

6. Review Comments to the Author

Reviewer #1: I thank the authors for their extensive efforts at improving the manuscript, particularly because their improvements will make their work much more approachable for teams looking for QI approaches for the IBD population under their care.

7. PLOS authors have the option to publish the peer review history of their article (what does this mean?). If published, this will include your full peer review and any attached files.

Reviewer #1: No

---

## [Editor Report · Acceptance letter]

26 Feb 2024

PONE-D-23-29768R1 

PLOS ONE

Dear Dr. Ridsdale, 

I'm pleased to inform you that your manuscript has been deemed suitable for publication in PLOS ONE. Congratulations! Your manuscript is now being handed over to our production team.

Kind regards, 

on behalf of

PD Dr. Valérie Pittet 

Academic Editor

PLOS ONE